# Interfacial Polarization Control Engineering and Ferroelectric PZT/Graphene Heterostructure Integrated Application

**DOI:** 10.3390/nano14050432

**Published:** 2024-02-27

**Authors:** Kaixi Bi, Shuqi Han, Jialiang Chen, Xiaoxue Bi, Xiangyu Yang, Liya Niu, Linyu Mei

**Affiliations:** 1School of Semiconductors and Physics, North University of China, Taiyuan 030051, China; bikaixi@nuc.edu.cn; 2Key Laboratory of National Defense Science and Technology on Electronic Measurement, North University of China, Taiyuan 030051, China; s202106058@st.nuc.edu.cn (S.H.); chjialiang@163.com (J.C.); s202101034@st.nuc.edu.cn (X.B.); s2006132@st.nuc.edu.cn (L.N.); 3Zhejiang Dali Technology Co., Ltd., Hangzhou 310053, China; yangxiangyu@dali-tech.com

**Keywords:** graphene, ferroelectric film, polarized substrate, rectification

## Abstract

Integration and miniaturization are the inevitable trends in the development of electronic devices. PZT and graphene are typical ferroelectric and carbon-based materials, respectively, which have been widely used in various fields. Achieving high-quality PZT/graphene heterogeneous integration and systematically studying its electrical properties is of great significance. In this work, we reported the characterization of a PZT film based on the sol–gel method. Additionally, the thickness of the PZT film was pushed to the limit size (~100 nm) by optimizing the process. The test results, including the remnant and leakage current, show that the PZT film is a reliable and suitable platform for further graphene-integrated applications. The non-destructive regulation of the electrical properties of graphene has been studied based on a domain-polarized substrate and strain-polarized substrate. The domain structures in the PZT film exhibit different geometric structures with ~0.3 V surface potential. The I–V output curves of graphene integrated on the surface of the PZT film exhibited obvious rectification characteristics because of p/n-doping tuned by an interfacial polarized electric field. In contrast, a ~100 nm thick PZT film makes it easy to acquire a larger strain gradient for flexural potential. The tested results also show a rectification phenomenon, which is similar to domain polarization substrate regulation. Considering the difficulty of measuring the flexural potential, the work might provide a new approach to assessing the flexural polarized regulation effect. A thinner ferroelectric film/graphene heterojunction and the polarized regulation of graphene will provide a platform for promoting low-dimension film-integrated applications.

## 1. Introduction

Graphene can be easily transferred onto the surface of specialized substrates to construct low-dimension film-integrated devices like photoelectric detectors, thin film transistors, and flexible electronic equipment [1,2,3,4]. Abundant physical and chemical mechanisms are contained at the interface between graphene and a functional substrate. In these applications, the Fermi level regulation or p/n doping of graphene has played a vital role in the final response output signal of devices. Benefiting from its large ratio of surface area to volume, the change in the dielectric environment at the interface can significantly influence the electric properties of graphene. So, a large number of works have regulated the electrical properties of graphene through the use of various methods, including chemical doping, high-energy particle beam irradiation, structural regulation and so on. Among these methods, chemical modification is a common approach for altering the electrical output signals of graphene. For usual chemically modified materials, inorganic molecules (NO_2_ and NH_3_) and organic molecules (nonaromatic or aromatic molecules) have been used as holes and electron donors to change the Fermi level of graphene [5,6,7]. Electrochemical intercalation based on ion liquids is another important chemical modification approach [8,9,10]. With the help of interlayer electrochemical reaction mechanisms, it has a stronger ability to change the electrical output signals of multilayer graphene film. However, the use of these methods usually leads to impurities, poor controllability and device performance degradation. Compared with chemical modification methods, high-energy particle beam irradiation technology is more commonly used for thick graphene modulation [11,12,13,14]. In the process, a high-energy particle beam is injected into a graphene crystal to change the atomic composition and energy band structure of graphene, which could destroy the perfect lattice of graphene. External field control methods can avoid damage to the crystal structure of graphene. Combined with the local back gate, a high-resolution modified location can be realized across multiple cycles [15,16,17]. Complex steps and higher costs are major challenges for further integration and promotion. Based on the above analysis, exploring non-destructive and precise regulation methods is of significance for the further development of graphene devices. 

Ferroelectric thin films have prominent polarization hysteresis, large dielectric constants, and electro-optical effects [18,19,20,21,22]. By designing a specific thin film structure or applying an excitation electric field, surface polarization potential will form, changing the properties of the interface film. Manipulating the interfacial polarization of ferroelectric films has become a research hotspot. The domain is the basic unit of ferroelectric materials. Many researchers have directly written domain arrays in films using a PFM probe or other polarization installations [23,24,25]. The structure size of the domain can reach the micro/nanometer scale. Its surface potential is usually several hundred millivolts. In addition, the surface polarization potential of ferroelectric films can also be acquired based on the flexoelectric effect. This effect refers to the electric polarization induced by the strain gradient, which was first proposed by Mashkevich and Tolpygo in 1957 [26,27]. Lee et al. at Seoul National University have deposited a 10 nm thick HoMnO_3_ ferroelectric film based on molecular beam epitaxy equipment [28]. Its flexural potential has been significantly improved due to a six orders of magnitude increase in the film strain gradient. The surface flexure potential can be further enhanced because of its large dielectric constant. As a result of the flexural enhancement effect of the ferroelectric thin film at a micro/nanometer scale, low-dimension materials on its surface will be tuned. Professor Liu at Xi’an Jiaotong University integrated a graphene/PZT composite film to change the concentration and types of carriers of a graphene crystal [29]. The results displayed linear regulation between flexural potential and the output signal of graphene. Qin Professor et al. at Lanzhou University have also taken advantage of flexural polarization charge to control the electrical transport of carriers at the interface as gate electrodes [30]. Currently, electron devices are developing in the direction of miniaturization and integration. Combined with the rise and application of new low-dimensional materials, there is a growing requirement to study the interface control of ferroelectric films and low-dimensional graphene films.

In this study, we have systematically studied the carrier transport behavior of graphene tuned by ferroelectric film based on interfacial polarization effect. Through the sol–gel method, a PZT film with micro/nanometer thickness was synthesized on Si/SiO_2_ and mica substrate. XRD characterization verifies the crystallization of PZT film with the perovskite phase. The corresponding diameter of the PZT nanometer is about 36.19 nm. Then, AFM and SEM images displayed smooth and dense film with 3.1 nm longitudinal roughness. The remnant polarization (Pr), coercive electric field (Ec), and leakage current of the PZT sample are ~69 μC/cm^2^, ~11.48 kV/mm, and 2.6 µA/cm^2^ at a ~540 kV/cm electric field intensity, respectively. The electrical properties indicate that PZT film can be used as reliable polarization platform. The domain structure in PZT film leads to ~0.3 V surface potential and obvious rectification characteristics, which verify the tunability of the polarized ferroelectric substrate. Additionally, the I–V output curves of graphene on the surface of a 100 nm thick PZT film were also tested under both bent and unbent states. With the production of external forces and generation of strain gradients, graphene film on its surface also demonstrated rectification characteristics on account of the interface polarization mechanism. PZT and graphene are important ferroelectric and carbon-based materials; exploring the interfacial regulatory relationship between PZT and graphene will be crucial to multilayer integrated applications.

## 2. Experiment

### 2.1. PZT Preparation

The solutes used in the precursor solution for PZT film preparation are Pb(CH_3_COO)_2_·3H_2_O, Zr(CH_3_CH_2_CH_2_O)_4_ and Ti(C_3_H_7_O)_4_. Acetic acid solvent (CH_3_COOH) was used to dissolve the above solutes. The atomic ratio of Pb:Zr:Ti was set as 1:0.3:0.7 by controlling the solution concentration. In the following experiments, ~5.0324 g Pb(CH_3_COO)_2_·3H_2_O was firstly dissolved in 15 mL of acetic acid solvent under 110 °C to acquire a 10 mL solution. Then, ~2.9763 g Zr(CH_3_CH_2_CH_2_O)_4_ and ~1.6528 g Ti(C_3_H_7_O)_4_ solute was added into Pb(CH_3_COO)_2_·3H_2_O solution, which was then cooled to 25 °C. Afterwards, we dropped 5 mL distilled water into a mixed solution for 30 min stirring on a constant temperature magnetic stirring table. Then, 2 mL lactic acid and 3 mL glycol were put into the solution, respectively, and 20 min of magnetic stirring was performed afterwards. By mixing with glycol ether and acetic acid in equal volumes, a lead acetate trihydrate solution with 0.4 mol/L was prepared. Finally, the solution was filtered and stood for 24 h to acquire PZT glue solution for subsequent PZT film preparation and polarization interface application.

In our work, the PZT/graphene composite structure was fabricated according to the following process. The sol–gel method was adopted to prepare PZT crystal film with different thicknesses varying from nanometer to the micron scale by controlling the spin-coating speed and solution concentration of the PZT sol solution. Generally, it is easier to obtain uniform and high-quality ferroelectric films with a small size sample. Considering the specific application requirements, PZT films could be deposited on the surface of the Pt/Ti substrate or mica sheet. Then, the sample was subjected to the following annealing treatment: (1) keeping the spin-coating film at 350 °C for 5 min is sufficient for organic materials, while repeating the process can acquire a thicker PZT film; (2) the crystallization of PZT film can be realized after sintering at 650 °C for 10 min.

### 2.2. Device Fabrication

After finishing the preparation of the PZT film, the graphene was transferred onto the surface of the sample. The graphene film used here was few-layer (2~3 layers) graphene prepared on the surface of a copper substrate by the chemical vapor deposition method. Compared with monolayer graphene, few-layer graphene has more stable electrical transport properties. In order to construct the graphene/PZT composite structure, a series of graphene transfer processes were carried out as follows. (1) First, PMMA film was spin-coated on the surface of graphene film. (2) Then, the sample was infiltrated into a mixed solution consisting of copper sulfate pentahydrate and diluted hydrochloric acid for 40 min. Deionized water also was used to clean the residual impurity ions of the sample. (3) When the copper substrate was etched and cleaned, PMMA/graphene film was transferred onto the surface of the PZT sample. By using an acetone solution to remove PMMA resist, graphene film was retained. Before transferring graphene, we first provided oxygen plasma treatment to enhance the adhesion force. 

Then, lift-off and E-beam evaporation processes were also executed to obtain a top electrode with a micron interval on the surface of the PZT/graphene composite film. Three types of metal film were adopted in the device fabrication process. For graphene/PZT film on a SiO_2_/Si substrate, thinner Ti/Au film was deposited on the surface of the PZT film to meet the basic electrical conductivity requirement. An impulse voltage was applied to the PZT interlayer between top probe and bottom metal layer. Arbitrary domain structures can be excited by the motion of the probe (Kelvin probe force microscopy). For graphene/PZT film on a mica substrate, thicker metal electrodes were needed to ensure good electrical conductivity in a large deformation state. The copper film deposition process based on magnetron sputtering has the advantages of low cost and fine ductility; then, 300 μm thick copper electrode was deposited on the surface of the PZT/mica structure. The bending degree of the PZT/graphene film can be precisely controlled by the external mechanical installation (Instron 5900, Instron, Norwood, MA, USA).

## 3. Results and Discussion

To investigate the crystallization of PZT film prepared by the sol–gel method, the XRD spectrum was measured for further material analysis [31,32,33]. Figure 1a shows the X-ray diffraction (XRD) pattern of a 1 µm thickness PZT film for subsequent polarization study. The characteristic peaks including (100), (110), (200), (121), (−121), and (−211) appeared in the test pattern, which verified the formation of the perovskite phase. The crystallization size was calculated based on the Scherrer–Debye formula: D = Kλ/βcosθ, in which K is constant, λ is the diffraction wavelength, β is the “full width at half maximum” of the peak, and θ is the angle measured. The (110) peak located at about 31.6° was usually used for sample calculation. The corresponding diameter value of PZT nanoparticles is ~36.19 nm. The lattice strain is another important parameter, which can also be calculated based on the formula: ε = βcosθ/4. The calculated value is 0.0011, which is smaller than the lattice strain parameter reported in a preliminary study. The results indicated that a smaller particle size and lattice strain are conducive to regulating the ferroelectric polarization properties.

The surface roughness and morphology of PZT film are important factors for low-dimensional material integration or on-chip TFT (thin film transistor) device applications. An AFM probe has a high-resolution response to the surface morphology and roughness. Figure 1b presents an image mapping of the selected PZT area with 20 µm × 20 µm size. The maximum height roughness R_max_ and mean surface roughness R_aver_ are 3.1 nm and 1.7 nm, respectively. A relatively smooth interface and subsequent oxygen plasma treatment can ensure high-quality low-dimensional integration. The thickness of PZT films on the Si/SiO_2_/Pt/Ti substrate and mica slice can be confirmed by SEM cross-section imaging characterization. Just as shown in Figure 1c,d, the PZT film is dense and uniform. The size of particles contained in the PZT film ranges from 30 to 40 nm, which is consistent with the XRD calculated results. The cross-section test sample was constructed by the laser-cutting or scissor-cutting method. Different functional layers exhibited good adhesion ability at the interface. These excellent structure results help improve the interface control ability of multilayer integrated materials and devices.

Studies of polarization properties and the local piezoresponse of PZT film based on the sol–gel method were tested by Piezoresponse Force Microscope equipment. The polarization vs. electric field (P–E) hysteresis loops were tested at room temperature (Figure 2) with an applied electric field intensity from −674 to 674 kV/cm. The corresponding remnant polarization (Pr) and coercive electric field (E_c_) values are ~69 μC/cm^2^ and ~11.48 kV/mm for the PZT film. As a typical saturated hysteresis loop, the results verify an obvious ferroelectric feature [34,35].

Generally, its surface potential can be regulated by adopting E_c_ three times to acquire an electric domain structure in a ferroelectric film. As an important ferroelectric film with interface polarization function, the leakage current of the PZT sample was ~ 2.6 µA/cm^2^ at ~540 kV/cm electric field intensity (as shown in Figure 3). The results exhibit impressive insulation properties, which can ensure a lower charge perturbation for further TFT (thin film transistor) integrated applications [36,37].

Figure 4 shows the capacitance-voltage (C–V) curve at both forward and reverse sweep voltage to study the PZT film. Two distinct two peaks were displayed (“butterfly” shape) because of polarization switching in 370 nm thick PZT film at ~405 kV/cm electric field intensity. The highest and lowest capacitances are ~240 pF and ~680 pF, respectively. By the plate capacitance formula of C = εs/4πkd (d ≈ 370 nm, k ≈ 8.99 N·m^2^/C^2^), the calculated permittivity is ~1.45 × 10^10^ F/m, which leads to higher flexural polarization potential when subjected to stress and deformation [38,39].

Polarization reversal of the ferroelectric domain has a huge influence on the interface physics. The direction of polarization can be tuned by changing the domain structure under a 10 V alternating voltage with a frequency of 1000 Hz. Benefiting from a small-size probe with 25 nm, the feature size of the domain can be written reliably in the micrometer or sub-micrometer range scale. Three kinds of domain images including a square, triangle, and circle (“bright” and “dark” regions) were obtained before and after applying DC voltage in the PZT film. The surface potential of domain structures was tested by KPFM (Kelvin probe force microscopy) probe mode. The tested potential difference is about 0.3 V at the domain boundary area between the up and down domain section. The domain/potential difference boundary in Figure 5 is not sharp enough because of some defects such as dislocations, grain boundaries, and lattice misfits, but it still can regulate the surface potential for low-dimensional material on a micrometer scale without causing any defects, which is important to change the energy band structure of the low-dimension film attached to its surface. 

The graphene/PZT on the SiO_2_/Si substrate is shown in Figure 6a. Ti/Au electrodes were deposited by the electron beam evaporation method. Graphene with a ~1 mm × 1 mm size was transferred onto the electrode gap with 10 μm intervals. Different oriented domain structures can be written by a PFM probe with the external electric field. An obvious dividing line can be found in the out-of-plane phase picture of the graphene/PZT sample, which means different polarized electric fields were formed to tune the carrier density of graphene. For the upper half region, electrons may be bound to the surface of the positive electric field, and more hole carriers were used for transport (p-doping). For the bottom half region, holes may be bound to the surface of the negative electric field, and more electrons were used for transport (n-doping). Raman tests have also been used to systematically characterize the properties of graphene. Figure 6c exhibits the Raman spectra of the graphene/PZT sample and pure graphene film. The peaks at 138, 197, 274, and 544 cm^−1^ come from the A1(1TO), E(2TO), B1 + E and E(3TO) mode. The spectrum verifies the crystallization of PZT film after the annealing process. Three typical characteristic peaks are located at the D peak (1380 cm^−1^), G peak (1582 cm^−1^), and 2D peak (2700 cm^−1^). The G-peak position represents the in-plane vibration pattern of sp2 hybrid carbon atoms in graphene, which is sensitive to the thickness of graphene. In Figure 6c, the G-peak at 1587 cm^−1^ indicates that the graphene used here is a few layers. The D-peak at ~1355 cm^−1^ means that the defects and disorder may be produced because of polarized PZT film. Generally, single-layer graphene has a 2D-peak Raman signal with ~30 cm^−1^ FWHM value and Gaussian distribution. In this work, the 2D-peak has presented a ~56 cm^−1^ FWHM (full width at half maximum) value. It illustrates that the graphene used here is a typical few-layer graphene sample. 

Combining with a follow-up lift-off and metal film deposition process, a graphene/PZT-integrated film transistor was fabricated (Figure 6d). The domain structure can be written by a PFM probe between two electrodes on the surface of PZT film. The opposite domain direction can lead to surface potential (~0.3 V) in our film sample. Up/down with different phases can further regulate graphene in the channel to construct the p-doping and n-doping of graphene. The current output curve of pure graphene was a symmetrical distribution with the bias voltage changing from −1 to 1 V. When graphene (Kelvin probe force microscopy) was placed on the surface of PZT film containing a domain structure with opposite orientation (Figure 6e), the domain orientation of the PZT film result in an opposite external electric field. Polarization of the electric field will lead to positive/negative charges and Fermi-level change. Then, the graphene sample has finished the conversion of the p-type and n-type. With the hole and electron movement in the p-type and n-type area of the graphene sample, a built-in electric field was produced in the graphene film. The p-n junction only allows the current to flow in one direction, so a slight rectification characteristic curve is presented in Figure 6e. In addition, the perovskite crystal type of PZT film based on Raman and XRD characterization implies a huge application prospect in the field of photoelectric integrated devices.

The flexoelectric effect plays an important role in the study of highly sensitive mechanical sensors. It is positively correlated with the dielectric parameters. The graphene/PZT on the mica substrate sample was fabricated and provided in Figure 7a. A thicker copper electrode with 300 μm thickness was prepared by magnetron sputtering equipment. CVD graphene was coated on both sides of PZT film based on a wet transfer process. Then, the composite structure can be placed on a fixing bracket. Mechanical deformation will be produced when subjected to external pressure, and strain can slightly change the domain orientation. The calculated bending radius was ~9.6 mm and ~7.8 mm in the electrical test process. The strain gradient may induce the change in domain polarization orientation. Then, p/n-doping will lead to a weak p-n junction inside the graphene film. Comparing the I–V curve of graphene/PZT on the Si-based substrate and mica substrate, we can find that strain gradient-induced surface potential change is weaker than the PFM probe modulation method.

A detailed schematic diagram of the graphene/PZT film is displayed in Figure 7b. When the PZT film is in a bent state, the strain gradient can produce surface potential and electric charge. We have provided the I–V electrical output curve of the graphene/mica composite structure in Figure 7b,c as a comparison. Graphene attached to the PZT film will be regulated by an interface electrical signal. PZT films with ~1 µm and ~100 nm thickness were adopted to integrate the PZT/graphene film device. Figure 7c corresponds to a PZT film with ~1 µm thickness; the I–V output curves at the original and different bending states were presented, in which the current is linearly dependent on voltage. Graphene has a higher current output value at its original state and a lower current at the stressing state. The results can be explained by an energy band change resulting from graphene deformation and external charge perturbation at the interface. For thinner PZT film with ~100 nm thickness, the I–V output current can also be provided in Figure 7d.

The graphene sample in the absence of the PZT film exhibits a smaller current value with the structure bending, which may result from the larger resistance value when subjected to external strain. Then, the graphene/PZT structure will generate a larger current value because of the surface polarization potential and charge injection. A bidirectional current was produced in the original state with external voltage varying from −1 to 1 V. The slight rectification characteristic curve appeared in the bending state, which means p/n-doping was produced from the PZT flexural polarization. Generally, ferroelectric materials have large dielectric constants. When the structure size reaches the nanometer scale, the strain gradient of ferroelectric materials will be greatly enhanced, which leads to huge flexoelectric charges and a potential to tune graphene on its surface. Considering the PZT is fragile in thicker film, a larger deformation of thicker PZT film with 1 μm thickness may result in the fragmentation of the membrane, which makes it difficult to produce surface polarization when subjected to external force. Then, the rectification properties can be found in Figure 7d rather than Figure 7c. In this case, graphene was divided into a p-type and n-type area for the built-in electric field. Then, a slight rectifier output electrical curve was formed in Figure 7d.

## 4. Conclusions

In summary, we have provided two kinds of polarized substrates to regulate the electrical properties of graphene. The polarized substrate was firstly constructed based on the domain structure, and ~0.3 V voltage was presented on the surface of the PZT film. The I–V curves of graphene on surface of the substrate exhibit obvious rectification characteristics. The high reliability of the spatial position and manufacturing accuracy of the domain structure provided an effective method for the precise regulation of the electrical properties of graphene. Strain polarization regulation was also an important way to construct a polarized substrate, because thinner PZT films are useful for improving the bending degree, and the enhanced bending degree will lead to a larger strain gradient and flexural polarization potential. Thus, a 100 nm thick PZT film has been used for flexural potential regulation. The results show that the carrier transport of graphene on the surface of the PZT film under an external strain gradient has also presented slight p/n-doping and a rectifier output. The results indicate that the ~100 nm PZT film has a similar polarization regulation ability based on flexural potential compared to traditional domain polarized regulation. All the systematic research will provide significant support in low-dimension functional film-integrated applications in various devices. 

## Figures and Tables

**Figure 1 nanomaterials-14-00432-f001:**
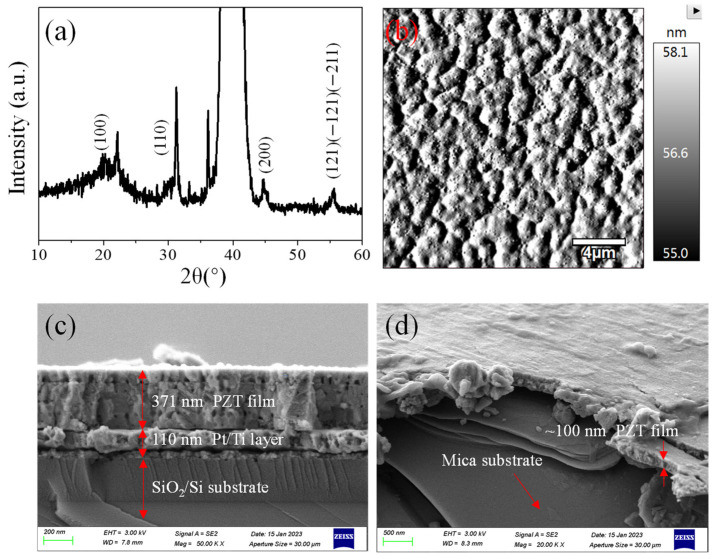
(**a**) XRD spectrum of the PZT film prepared under different temperatures; (**b**) topographical image of PZT film with 20 µm × 20 µm scanning area characterized by AFM equipment; (**c**,**d**) SEM cross-section images of PZT film deposited on Si/SiO_2_/Pt/Ti and mica substrate.

**Figure 2 nanomaterials-14-00432-f002:**
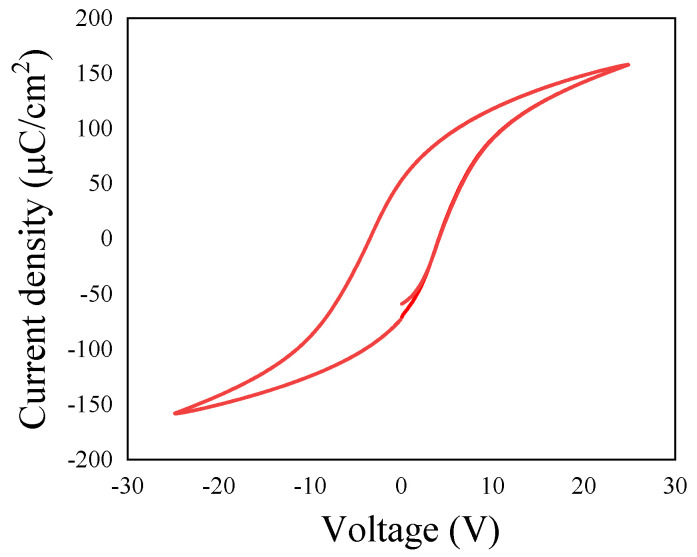
P–E hysteresis loops of the PZT film with excitation voltage from −25 to 25 V.

**Figure 3 nanomaterials-14-00432-f003:**
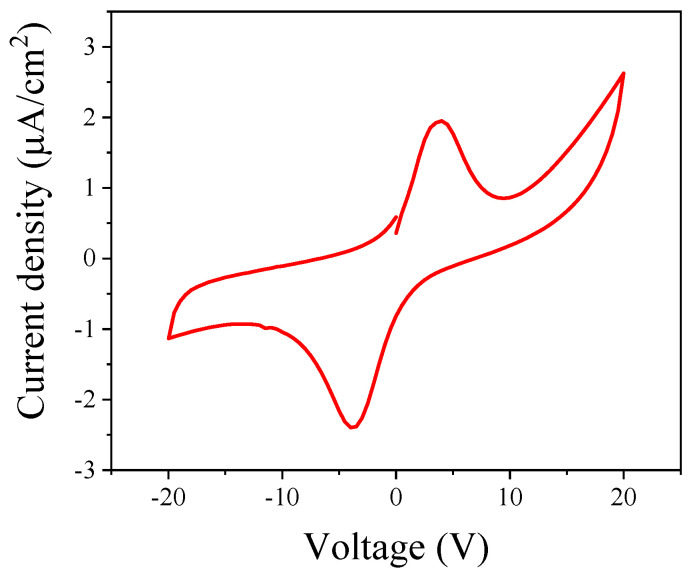
Leakage current based on the same measure equipment and sample.

**Figure 4 nanomaterials-14-00432-f004:**
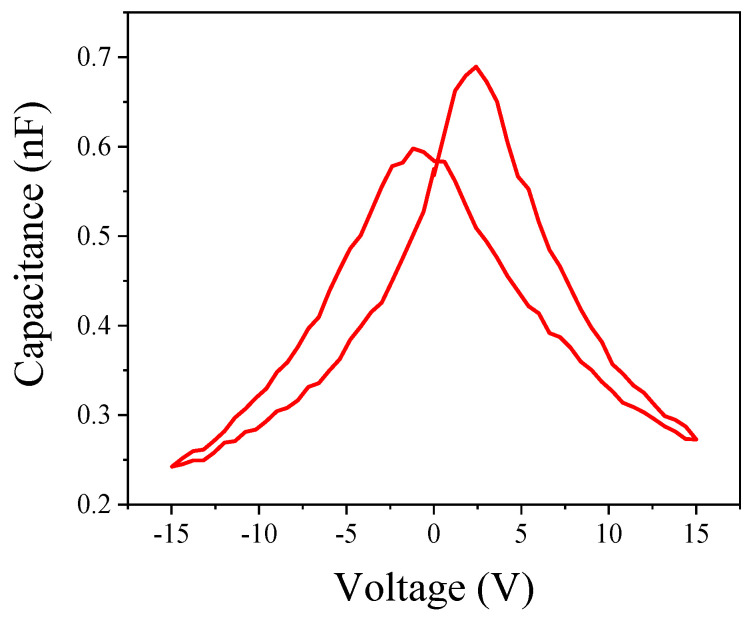
Capacitance–voltage characteristic tested by PFM probe on the surface of Pt/Ti/PZT film.

**Figure 5 nanomaterials-14-00432-f005:**
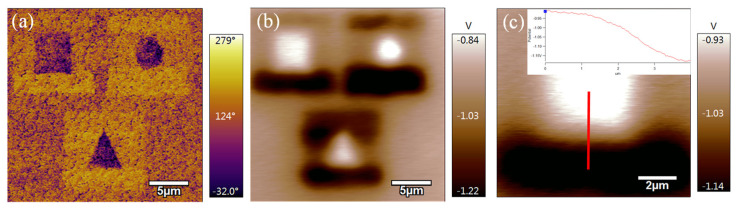
(**a**) Image of ferroelectric domains written by PFM probe with 10 V tip voltage on the PZT film surface. Three domain structures present a ~180° contrast in phase test results; (**b**,**c**) surface potential of thin film was characterized by KFPM probe in 20 × 20 µm^2^ area. The potential value at the boundary is ~0.3 V.

**Figure 6 nanomaterials-14-00432-f006:**
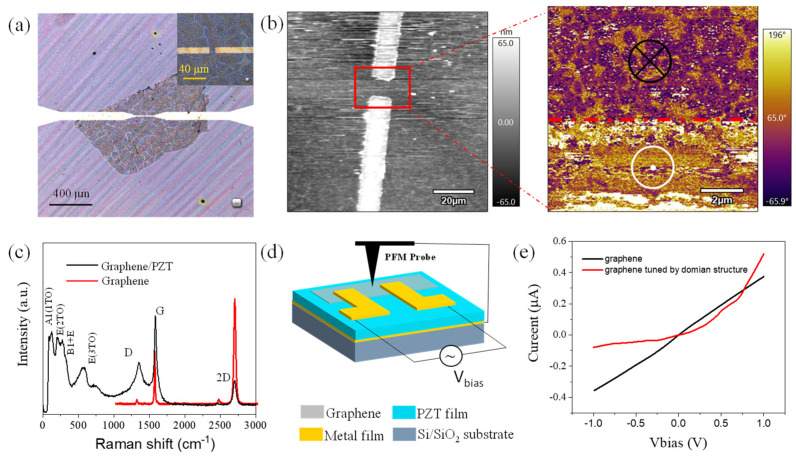
(**a**) Photograph of graphene/PZT sample on SiO_2_/Si substrate. The upper right image is a partial enlargement of the electrode gap area. (**b**) The gray picture displayed the structure topography based on PFM (piezo-response force microscopy) equipment. The yellow picture shows the domain structure characterization of the middle region of the electrode. (**c**) Raman spectrum of pure graphene and graphene/PZT integrated film; (**d**) Test and structure schematic diagram of graphene/PZT device; (**e**) I–V output electrical transport test curve.

**Figure 7 nanomaterials-14-00432-f007:**
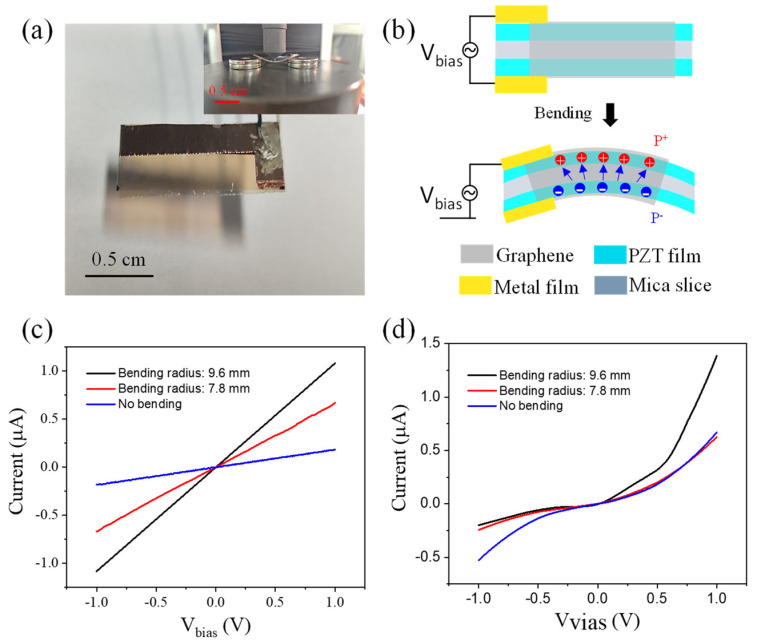
(**a**) Photograph of graphene/PZT sample on mica substrate. Copper electrode with 300 μm thickness was deposited on surface of PZT film by magnetron sputtering equipment. The upper right image is a sample image under bending condition. (**b**) Working diagram of graphene/PZT device; (**c**,**d**) I–V output electrical transport test curve of 1 µm and 100 nm thickness, respectively.

## Data Availability

The research data of this work are available upon request to authors.

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
