# Peer review of "Interfacial Polarization Control Engineering and Ferroelectric PZT/Graphene Heterostructure Integrated Application"

_nanomaterials, 2024, doi:10.3390/nano14050432_

Round 1

Reviewer 1 Report (New Reviewer)

Comments and Suggestions for Authors

In this study, ferroelectric PZT and graphene heterostructure have been fabricated.

1. What is the minimum thickness of PZT 100 nm or 110 nm? Please check line number 13 (abstract section) and line number 99 ( introduction section).

2. The I-V output curves exhibit obvious rectification characterization and p/n-doping. Please correct this sentence.

3. SEM was used to measure the thickness of the film deposited on the substrate.  "Then AFM and SEM images displayed smooth and dense granular film with 3.1 nm height 91 roughness and 1.7 nm mean surface roughness". Correct this sentence.

4. As PZT is hazardous, the author should explain the reason for selecting lead-based ferroelectric materials.

5. Line numbers 106-147 should be included in the experimental section.

6. Two types of substrates were used for developing PZT films why?

7. The XRD and P-E hysteresis properties provided here are of which film developed on Si/SiO2 substrate or Mica? any effect of the thickness on these properties.

8. Line number 304 have the authors analyzed dielectric properties of the PZT films developed in this research work.

Author Response

Response Letter

Academic Editor Notes

The manuscript requires extensive editing to rectify its English. In fact, a complete rewrite is necessary. There are numerous grammar errors and incorrect word usage in the text, certain sentences resemble fragments, and many are very challenging to comprehend. However, it is worth noting that the authors have addressed the scientific and technical comments provided by the referees.

Response:

Thank you for your careful and professional reviewing. Here, we have systematically studied the carrier transport behavior of graphene tuned by ferroelectric film based on interfacial polarization effect. The performance of domain polarization modulation and flexural polarization modulation have been compared in the same ferroelectric material and preparation process. Furthermore, the thickness of PZT film also has been almost pushed into the thinnest size. Relevant theories and techniques are of great reference value for the integrated application of on-chip ferroelectric/graphene devices. It is a very important problem that the English expression mentioned by editor and reviewer. We have checked language expression through the whole manuscript. The modified parts were marked with bright yellow background. Thank you again for your positive comments.

Reviewer #1:

Comments:

In this study, ferroelectric PZT and graphene heterostructure have been fabricated.

Response:

The dynamic tunability of graphene’s electrical properties is a huge advantage and research hotspot. As you said, in this paper, PZT/ graphene heterojunction was constructed based on the research strategy of interfacial polarization modulation method. The electrical transport behavior of graphene has been tuned and analyzed by two interfacial polarization modes of PZT ferroelectric thin film electric domain surface potential and flexor potential. Electrical test curves of graphene tuned by two methods exhibited relatively obvious rectifier output features. This study provides theoretical and technical support for the further research of on-chip integrated ferroelectric/graphene detector and flexible electro-mechanical coupled transducer.

Comments:

  1. What is the minimum thickness of PZT 100 nm or 110 nm? Please check line number 13 (abstract section) and line number 99 (introduction section).

Response:

Thanks for your careful reviewing. The thinnest thickness of PZT film prepared by sol-gel method is ~ 100nm. The 110 nm thickness in line 99 is incorrect. We have corrected the thickness of 110nm in line 99 to 100nm, and the relevant description has also been checked and modified in the manuscript.

Comments:

  1. The I-V output curves exhibit obvious rectification characterization and p/n-doping. Please correct this sentence.

Response:

Thanks for your professional review and constructive suggestions. We have rechecked and corrected the sentence as follows:

The I-V output curves of graphene integrated on surface of PZT film exhibited obvious rectification characteristic because of p/n-doping tuned by interfacial polarized electric field.

We have also modified it in line 19 (abstract section).

Comments:

  1. SEM was used to measure the thickness of the film deposited on the substrate.  "Then AFM and SEM images displayed smooth and dense granular film with 3.1 nm height 91 roughness and 1.7 nm mean surface roughness". Correct this sentence.

Response:

Thanks for your professional review and constructive suggestions. We have rechecked and corrected the sentences as follows:

Then AFM and SEM images displayed smooth and dense film with 3.1 nm longitudinal roughness.

We have also modified it in line 94 (introduction section).

Comments:

  1. As PZT is hazardous, the author should explain the reason for selecting lead-based ferroelectric materials.

Response:

Thanks for your professional reviewing. The problem you mentioned does exist. It is precisely because ferroelectric materials containing lead are dangerous and have environmental pollution, and many researchers tend to study some green lead-free ferroelectric materials (Science, 2023, 380, 1034–1038; Journal of Materials Chemistry C, 2020, 10, 1039; ACS Applied electronic Materials, 2022, 4(5), 2109-2145). Although PZT ferroelectric film used in the manuscript is not cutting-edge non-lead materials, it still has some significant advantages such as low-cost, large residual polarization, large dielectric constant and low crystallization temperature. The more important thing is that the fabrication process of PZT film is easy to be compatible with the integrated manufacturing process of other electronic devices. As an typical ferroelectric material, the study of ferroelectric properties of PZT is still of great significance (Sensors and Actuators A: Physical, 2023, 349(1), 114020; Microsystems & Nanoengineering, 2022, 122; Materials today, 2021, 43(1), 407-412; AIP advances, 2024, 14, 025213).

Comments:

  1. Line numbers 106-147 should be included in the experimental section.

Response:

Thanks for your valuable reviewing. The contents in line numbers 106-107 mainly about materials and device preparation. We have made corresponding changes in the text, see section Experiment section.

Comments:

  1. Two types of substrates were used for developing PZT films why?

Response:

Thanks for your reviewing. In this paper, two methods for generating interfacial polarization field of PZT ferroelectric thin films are proposed: (1) The first sample is a PZT film deposited on Pt/Ti/SiO2/Si substrate. The polarization orientation of domain in PZT film was mainly modulated by external electrical field applied between the top PFM probe and bottom metal layer. Then interface polarization voltage in PZT film with opposite direction is generated to achieve the modulation of the electrical properties of graphene integrated on surface of PZT film. In this mode, a metal layer with a silicon substrate is required to generate an external electric field for PZT polarization modulation. (2) The second sample is a PZT film deposited on flexible mica substrate based on sol-gel technology. Flexible substrate is useful to support PZT film at large strain gradient and bending state. According to flexoelectric theory, the surface flexural polarization potential of PZT ferroelectric thin film can be produced to modulate the surface graphene when subjected to external strain. In this mode, only mica flexible substrate can support strain modulation behavior.

Comments:

  1. The XRD and P-E hysteresis properties provided here are of which film developed on Si/SiO2 substrate or Mica? any effect of the thickness on these properties.

Response:

Thanks for your careful reviewing. The XRD and P-E hysteresis properties provided here have been acquired from PZT film on Si/SiO2 substrate. The single layer PZT film can be prepared based on sol-gel method mentioned in manuscript. Different thickness PZT film can be obtained by repeating the same steps. For thicker PZT film prepared by more coating cycles, larger grains and a few voids may be produced. In thee preparation process, the PZT crystal was firstly formed start from bottom, then subsequent spin-coating precursors will gradually grow around the bottom nucleation center. Due to the same annealing temperature and precursors were used in PZT preparation process, the crystallization properties of PZT will not change significantly except for the strength enhancement. The P-E hysteresis loop is an important characteristic parameter of ferroelectric materials. The polarization intensity of ferroelectric materials is not linear with the electric field, but shows a hysteresis relationship. Generally, the better symmetry can be observed in thicker PZT film because of weaker spontaneous responses. The remnant polarization value is a cumulant, then it will be increased with the increase of PZT film thickness. The coercive filed value in P-E hysteresis loop is a physical parameter related to the thickness of ferroelectric film. Although many researchers have studied the relationship between physical quantity related to the thickness of ferroelectric film and ferroelectric film thickness, no accurate calculation formula can be obtained. It can be found that coercive filed value will be increased with the decreasing of PZT film thickness. (Journal of the Korean Physical Society, 2011, 58(4), 809816; J Mater Sci: Mater Electron, 2013, 24:160–165; Thin Solid Films 2001,385, 5-10; Ferroelectrics, 2002, 271(1), 327-332).

Comments:

  1. Line number 304 have the authors analyzed dielectric properties of the PZT films developed in this research work.

Response:

Thanks for your careful and professional reviewing. PZT film has higher dielectric constant than conventional dielectric materials, which is important to acquire larger flexural potential. The flexural effect was firstly proposed by Mashkevich and Tolpygo of the Kyiv State University and the Kyiv Institute of Technology in 1957. This effect refers to the electrical polarization induced by the strain gradient. Tagantsev et al. of the Institute of Scientific Physics of the former Soviet Union conducted a systematic study on the value of flexoelectric response. The flexural potential can be calculated as following formula:  ( is flexural polarization intensity;  is the dielectric constant;  is strain gradient). It can be seen from above formula the flexural potential is positively related to the dielectric constant of ferroelectric materials. Generally, ferroelectric materials usually have a large dielectric constant, and the sharply increased stress gradient at the nanoscale can increase the flexoelectric effect by several orders of magnitude. So PZT ferroelectric film was adopted in this paper, and its dielectric constant is also briefly discussed. (Science, 2018, 360, 904-907; Physical Review B-Condens Matter, 1986, 34(8), 5883-5889; Journal of Applied Physics, 2013, 113, 194102)

Reviewer #2:

Comments:

By optimizing the sol-gel method, the authors were able to achieve a minimum thickness of the PZT film to be approx.  100 nm. A series of tests were made to evaluate the chemical or physical properties of the PZT film. The tests included measuring both the remnant voltage and the leakage current. The results showed that the carrier transport of graphene on the surface of the PZT film under external strain gradient have led to slight p/n-doping and rectifier output.

Response:

Thanks for your careful and accurate comments. Sol-gel method was used to prepare PZT film. By optimizing the sol-gel process, the thickness of PZT film also has been almost pushed into the thinnest size (~100 nm). Considering that the dynamic tunability of graphene’s electrical properties is a hot topic. Then graphene was integrated on the surface of PZT film in same fabrication process. We have also characterized the carrier transport curves of graphene tuned by ferroelectric film based on interfacial polarization effect. The performance of domain polarization modulation and flexural polarization modulation have been compared in the same ferroelectric material and preparation process. Relevant theories and techniques are of great reference value for the integrated application of on-chip ferroelectric/graphene devices.

Comments:

The language is very poor. I suggest the authors to rewrite Introduction and Conclusions in better English. I do not understand what they mean by the word ‘regulation’ i.e. polarized substrate regulation   and non-destructive regulation.

Response:

Thanks for your professional reviewing. We have rewrite Introduction and Conclusions of manuscript.

Ferroelectric materials have a non-centrosymmetric crystal structure, which will form an internal electric field in the ferroelectric body and induce the surface potential. This kind of substrate with a surface potential is called the polarization substrate in this paper, and the corresponding surface potential is ~ 0.3V. The electronic properties of graphene can be modulated by surface potential. Then we considered this process as polarized substrate regulation.

There are many ways to regulate the electrical properties of graphene, including chemical doping, high-energy particle beam irradiation modification, structural regulation and so on (Applied Physics Letters, 2011, 99, 033109; Advanced Materials Interfaces, 2020, 10, 1002; Nature Reviews Physics, 2021, 3, 791-82). Lots of attention has been payed to study the modulation methods, but these methods usually caused irreversible changes to the lattice structure or chemical bonds of graphene. As the result of that, there will be some degradation in the electrical performance of graphene. External electric field modulation method do not cause destructive to graphene crystal, and can be used repeatedly in multiple cycles. So two interfacial polarization modulation methods mentioned in the manuscript was called as non-destructive regulation.

Reviewer 2 Report (New Reviewer)

Comments and Suggestions for Authors

By optimizing the sol-gel method, the authors  were able to achieve a  minimum thickness of the PZT film to be approx.  100 nm. A series of tests were made to evaluate the chemical or physical properties of the PZT film. The tests included measuring both the remnant voltage and the leakage current. The results showed that the carrier transport of graphene on the surface of the  PZT film under external strain gradient have led to slight p/n-doping and rectifier output.

The language is very poor. I suggest the authors to rewrite Introduction and Conclusions in better English. I do not understand what they mean by the word ‘regulation’ i.e. polarized substrate regulation   and non-destructive regulation.

Comments on the Quality of English Language

The language is  bad.

Author Response

Response Letter

Academic Editor Notes

The manuscript requires extensive editing to rectify its English. In fact, a complete rewrite is necessary. There are numerous grammar errors and incorrect word usage in the text, certain sentences resemble fragments, and many are very challenging to comprehend. However, it is worth noting that the authors have addressed the scientific and technical comments provided by the referees.

Response:

Thank you for your careful and professional reviewing. Here, we have systematically studied the carrier transport behavior of graphene tuned by ferroelectric film based on interfacial polarization effect. The performance of domain polarization modulation and flexural polarization modulation have been compared in the same ferroelectric material and preparation process. Furthermore, the thickness of PZT film also has been almost pushed into the thinnest size. Relevant theories and techniques are of great reference value for the integrated application of on-chip ferroelectric/graphene devices. It is a very important problem that the English expression mentioned by editor and reviewer. We have checked language expression through the whole manuscript. The modified parts were marked with bright yellow background. Thank you again for your positive comments.

Reviewer #1:

Comments:

In this study, ferroelectric PZT and graphene heterostructure have been fabricated.

Response:

The dynamic tunability of graphene’s electrical properties is a huge advantage and research hotspot. As you said, in this paper, PZT/ graphene heterojunction was constructed based on the research strategy of interfacial polarization modulation method. The electrical transport behavior of graphene has been tuned and analyzed by two interfacial polarization modes of PZT ferroelectric thin film electric domain surface potential and flexor potential. Electrical test curves of graphene tuned by two methods exhibited relatively obvious rectifier output features. This study provides theoretical and technical support for the further research of on-chip integrated ferroelectric/graphene detector and flexible electro-mechanical coupled transducer.

Comments:

  1. What is the minimum thickness of PZT 100 nm or 110 nm? Please check line number 13 (abstract section) and line number 99 (introduction section).

Response:

Thanks for your careful reviewing. The thinnest thickness of PZT film prepared by sol-gel method is ~ 100nm. The 110 nm thickness in line 99 is incorrect. We have corrected the thickness of 110nm in line 99 to 100nm, and the relevant description has also been checked and modified in the manuscript.

Comments:

  1. The I-V output curves exhibit obvious rectification characterization and p/n-doping. Please correct this sentence.

Response:

Thanks for your professional review and constructive suggestions. We have rechecked and corrected the sentence as follows:

The I-V output curves of graphene integrated on surface of PZT film exhibited obvious rectification characteristic because of p/n-doping tuned by interfacial polarized electric field.

We have also modified it in line 19 (abstract section).

Comments:

  1. SEM was used to measure the thickness of the film deposited on the substrate.  "Then AFM and SEM images displayed smooth and dense granular film with 3.1 nm height 91 roughness and 1.7 nm mean surface roughness". Correct this sentence.

Response:

Thanks for your professional review and constructive suggestions. We have rechecked and corrected the sentences as follows:

Then AFM and SEM images displayed smooth and dense film with 3.1 nm longitudinal roughness.

We have also modified it in line 94 (introduction section).

Comments:

  1. As PZT is hazardous, the author should explain the reason for selecting lead-based ferroelectric materials.

Response:

Thanks for your professional reviewing. The problem you mentioned does exist. It is precisely because ferroelectric materials containing lead are dangerous and have environmental pollution, and many researchers tend to study some green lead-free ferroelectric materials (Science, 2023, 380, 1034–1038; Journal of Materials Chemistry C, 2020, 10, 1039; ACS Applied electronic Materials, 2022, 4(5), 2109-2145). Although PZT ferroelectric film used in the manuscript is not cutting-edge non-lead materials, it still has some significant advantages such as low-cost, large residual polarization, large dielectric constant and low crystallization temperature. The more important thing is that the fabrication process of PZT film is easy to be compatible with the integrated manufacturing process of other electronic devices. As an typical ferroelectric material, the study of ferroelectric properties of PZT is still of great significance (Sensors and Actuators A: Physical, 2023, 349(1), 114020; Microsystems & Nanoengineering, 2022, 122; Materials today, 2021, 43(1), 407-412; AIP advances, 2024, 14, 025213).

Comments:

  1. Line numbers 106-147 should be included in the experimental section.

Response:

Thanks for your valuable reviewing. The contents in line numbers 106-107 mainly about materials and device preparation. We have made corresponding changes in the text, see section Experiment section.

Comments:

  1. Two types of substrates were used for developing PZT films why?

Response:

Thanks for your reviewing. In this paper, two methods for generating interfacial polarization field of PZT ferroelectric thin films are proposed: (1) The first sample is a PZT film deposited on Pt/Ti/SiO2/Si substrate. The polarization orientation of domain in PZT film was mainly modulated by external electrical field applied between the top PFM probe and bottom metal layer. Then interface polarization voltage in PZT film with opposite direction is generated to achieve the modulation of the electrical properties of graphene integrated on surface of PZT film. In this mode, a metal layer with a silicon substrate is required to generate an external electric field for PZT polarization modulation. (2) The second sample is a PZT film deposited on flexible mica substrate based on sol-gel technology. Flexible substrate is useful to support PZT film at large strain gradient and bending state. According to flexoelectric theory, the surface flexural polarization potential of PZT ferroelectric thin film can be produced to modulate the surface graphene when subjected to external strain. In this mode, only mica flexible substrate can support strain modulation behavior.

Comments:

  1. The XRD and P-E hysteresis properties provided here are of which film developed on Si/SiO2 substrate or Mica? any effect of the thickness on these properties.

Response:

Thanks for your careful reviewing. The XRD and P-E hysteresis properties provided here have been acquired from PZT film on Si/SiO2 substrate. The single layer PZT film can be prepared based on sol-gel method mentioned in manuscript. Different thickness PZT film can be obtained by repeating the same steps. For thicker PZT film prepared by more coating cycles, larger grains and a few voids may be produced. In thee preparation process, the PZT crystal was firstly formed start from bottom, then subsequent spin-coating precursors will gradually grow around the bottom nucleation center. Due to the same annealing temperature and precursors were used in PZT preparation process, the crystallization properties of PZT will not change significantly except for the strength enhancement. The P-E hysteresis loop is an important characteristic parameter of ferroelectric materials. The polarization intensity of ferroelectric materials is not linear with the electric field, but shows a hysteresis relationship. Generally, the better symmetry can be observed in thicker PZT film because of weaker spontaneous responses. The remnant polarization value is a cumulant, then it will be increased with the increase of PZT film thickness. The coercive filed value in P-E hysteresis loop is a physical parameter related to the thickness of ferroelectric film. Although many researchers have studied the relationship between physical quantity related to the thickness of ferroelectric film and ferroelectric film thickness, no accurate calculation formula can be obtained. It can be found that coercive filed value will be increased with the decreasing of PZT film thickness. (Journal of the Korean Physical Society, 2011, 58(4), 809816; J Mater Sci: Mater Electron, 2013, 24:160–165; Thin Solid Films 2001,385, 5-10; Ferroelectrics, 2002, 271(1), 327-332).

Comments:

  1. Line number 304 have the authors analyzed dielectric properties of the PZT films developed in this research work.

Response:

Thanks for your careful and professional reviewing. PZT film has higher dielectric constant than conventional dielectric materials, which is important to acquire larger flexural potential. The flexural effect was firstly proposed by Mashkevich and Tolpygo of the Kyiv State University and the Kyiv Institute of Technology in 1957. This effect refers to the electrical polarization induced by the strain gradient. Tagantsev et al. of the Institute of Scientific Physics of the former Soviet Union conducted a systematic study on the value of flexoelectric response. The flexural potential can be calculated as following formula:  ( is flexural polarization intensity;  is the dielectric constant;  is strain gradient). It can be seen from above formula the flexural potential is positively related to the dielectric constant of ferroelectric materials. Generally, ferroelectric materials usually have a large dielectric constant, and the sharply increased stress gradient at the nanoscale can increase the flexoelectric effect by several orders of magnitude. So PZT ferroelectric film was adopted in this paper, and its dielectric constant is also briefly discussed. (Science, 2018, 360, 904-907; Physical Review B-Condens Matter, 1986, 34(8), 5883-5889; Journal of Applied Physics, 2013, 113, 194102)

Reviewer #2:

Comments:

By optimizing the sol-gel method, the authors were able to achieve a minimum thickness of the PZT film to be approx.  100 nm. A series of tests were made to evaluate the chemical or physical properties of the PZT film. The tests included measuring both the remnant voltage and the leakage current. The results showed that the carrier transport of graphene on the surface of the PZT film under external strain gradient have led to slight p/n-doping and rectifier output.

Response:

Thanks for your careful and accurate comments. Sol-gel method was used to prepare PZT film. By optimizing the sol-gel process, the thickness of PZT film also has been almost pushed into the thinnest size (~100 nm). Considering that the dynamic tunability of graphene’s electrical properties is a hot topic. Then graphene was integrated on the surface of PZT film in same fabrication process. We have also characterized the carrier transport curves of graphene tuned by ferroelectric film based on interfacial polarization effect. The performance of domain polarization modulation and flexural polarization modulation have been compared in the same ferroelectric material and preparation process. Relevant theories and techniques are of great reference value for the integrated application of on-chip ferroelectric/graphene devices.

Comments:

The language is very poor. I suggest the authors to rewrite Introduction and Conclusions in better English. I do not understand what they mean by the word ‘regulation’ i.e. polarized substrate regulation   and non-destructive regulation.

Response:

Thanks for your professional reviewing. We have rewrite Introduction and Conclusions of manuscript.

Ferroelectric materials have a non-centrosymmetric crystal structure, which will form an internal electric field in the ferroelectric body and induce the surface potential. This kind of substrate with a surface potential is called the polarization substrate in this paper, and the corresponding surface potential is ~ 0.3V. The electronic properties of graphene can be modulated by surface potential. Then we considered this process as polarized substrate regulation.

There are many ways to regulate the electrical properties of graphene, including chemical doping, high-energy particle beam irradiation modification, structural regulation and so on (Applied Physics Letters, 2011, 99, 033109; Advanced Materials Interfaces, 2020, 10, 1002; Nature Reviews Physics, 2021, 3, 791-82). Lots of attention has been payed to study the modulation methods, but these methods usually caused irreversible changes to the lattice structure or chemical bonds of graphene. As the result of that, there will be some degradation in the electrical performance of graphene. External electric field modulation method do not cause destructive to graphene crystal, and can be used repeatedly in multiple cycles. So two interfacial polarization modulation methods mentioned in the manuscript was called as non-destructive regulation.

This manuscript is a resubmission of an earlier submission. The following is a list of the peer review reports and author responses from that submission.

Round 1

Reviewer 1 Report

Comments and Suggestions for Authors

This paper shows multifunctional operation using PZT and graphene layer. And also, it shows possibility of flexible electronics using bending test. However, there are many content errors, typos, and problems with figures. Additionally, it is difficult to accepts the paper because it does not show any innovative parts.

1.     PZT sample shows Pr=0.3uC/cm2. Is is real? The figure shows different value.

2.     There is no proper explanation of the graphene properties and graphene transfer method in the mica/PZT structure.

3.     There is a lack of explanation for accurate rectifying of graphene's output curve.

4.     There is a lack of information about bending ratio in bending tests. Additionally, information in the absence of a PZT thin film is also lacking.

5.     Overall, there are problems with the English and there are many typos.

6.     Most of the content has already come from other papers, and there is nothing special about the paper itself.

Comments on the Quality of English Language

English is not good. It is recommended that you request an English proofreading agency.

Author Response

Response Letter

Reviewer #1: nanomaterials-2733135

Comments:

This paper shows multifunctional operation using PZT and graphene layer. And also, it shows possibility of flexible electronics using bending test. However, there are many content errors, typos, and problems with figures. Additionally, it is difficult to accepts the paper because it does not show any innovative parts.

Response:

Thanks for your professional comments. We have solved the problems you mentioned one by one.

Comments:

  1. PZT sample shows Pr=0.3uC/cm2. Is it real? The figure shows different value.

Response:

I am very sorry for the error. Remnant polarization is the value of polarization intensity given by the intersection of the ferroelectric hysteresis loop and the vertical axis. The remnant polarization (Pr) in figure 3 should be ~ 69 μC/cm2. The thickness of PZT tested here is ~371 nm. The corresponding coercive electric field strength can be acquired by calculating the ratio of coercive electric field to PZT thickness, i.e. ~11.48 kV/mm. The applied electric filed intensity was varied from -674 kV/cm to 674 kV/cm, which can be acquired by calculating the ration of the maximum voltage and thickness of PZT. The modified results was highlighted in manuscript.

We change “The polarization vs. electric field (P–E) hysteresis loops were tested at room temperature (figure 3) with an applied electric field intensity from -600 kV/cm to 600 kV/cm. Corresponding remnant polarization (Pr) and coercive electric field (Ec) values are ~ 0.3 μC/cm2 and ~5 kV/cm for PZT film.” to “The polarization vs. electric field (P–E) hysteresis loops were tested at room temperature (figure 3) with an applied electric field intensity from -674 kV/cm to 674 kV/cm. Corresponding remnant polarization (Pr) and coercive electric field (Ec) values are ~ 69 μC/cm2 and ~11.48 kV/mm for PZT film.”

Comments:

  1. There is no proper explanation of the graphene properties and graphene transfer method in the mica/PZT structure.

Response:

Thanks for your careful and rigorous review. Graphene sample used here was few-layer (2~3 layers) graphene prepared on surface of copper substrate by Chemical vapor deposition method. Compared with monolayer graphene, few-layer graphene has more stable electrical transport properties. In order to construct graphene/PZT composite structure, a series of graphene transfer processes were carried out as follows: (1) PMMA film was firstly spin-coated on the surface of graphene film; (2) Sample was infiltrated into mixed solution consisted of copper sulfate pentahydrate and diluted hydrochloric acid for 40 mins. Deionized water also was used to clean the residual impurity ions of sample; (3) When copper substrate was etched and cleaned, PMMA/graphene film was transferred onto surface of PZT sample. By using acetone solution to remove PMMA resist, graphene film was retained.

We have added the description in figure 1 “results and discussion” of manuscript.

Raman tests are also used to systematically characterize the properties of graphene. G-peak position represents the in-plane vibration pattern of sp2 hybrid carbon atoms in graphene, which is sensitive to the thickness of graphene. In figure 7, the G-peak at 1587 cm-1 indicate that the graphene used here is few layers. D-peak at ~1355 cm-1 means that the defects and disorder maybe produced because of polarized PZT film. Generally, single layer graphene has 2D-peak Raman signal with ~30 cm-1 FWHM value and Gaussian distribution. In this work, 2D-peak has presented ~ 56 cm-1 FWHM (full width at half maximum) value. It illustrates that graphene used here is a typical few-layer graphene sample.

We have added the description in figure 7 “results and discussion” of manuscript.

Comments:

  1. There is a lack of explanation for accurate rectifying of graphene's output curve.

Response:

Thanks for your professional comments. The current output curve of pure graphene was a symmetrical distribution with the bias voltage changes from -1V to 1V. When graphene was placed on the surface of PZT film containing domain structure with opposite orientation (figure 7), the domain orientation of PZT film will result in opposite external electric field. Polarization electric field will lead to positive/negative charges and fermi-level change. Then graphene sample has finished the conversion of p-type and n-type. With the hole and electron movement in p-type and n-type area of graphene, a built-in electric field was produced in graphene film. The p-n junction only allows the current to flow in one direction, so a slight rectification characteristic curve is presented in Figure 7e. (ACS Nano, 2021,15,7,10982-11013; Adv. Optical Mater. 2021, 9, 2100245)

Comments:

  1. There is a lack of information about bending ratio in bending tests. Additionally, information in the absence of a PZT thin film is also lacking.

Response:

Thanks for your professional comments. By referring to the relevant literatures, we find that bending radius maybe more commonly parameter in flexible device description. The length (L) and bending height (d) of graphene/PZT composite structure can be directly tested, and the corresponding bending radius can be calculated by Pythagorean Theorem. The calculation results show that the bending radius of bending 1 and bending 2 is ~ 9.6 mm and ~ 7.8 mm, respectively.

We also provide the I-V electrical output curve of graphene/mica composite structure in figure 8b and 8c as a comparison. The graphene sample in the absence of PZT film exhibit smaller current value with the structure bending, which maybe result from larger resistance value when subjected to external strain. Then graphene/PZT structure will generate larger current value because of surface polarization potential and charge injection. Comparing with 1 μm thick PZT film, 100 nm thick PZT suspended film will generate larger strain gradient under the same bending radius, which result in larger surface flexural potential and chargers. In the case, graphene was divided into p-type and n-type area for built-in electric field. So a slight rectifier output electrical curve was formed in figure 8d.

Figure 1 Calculation diagram of bending radius

Comments:

  1. Overall, there are problems with the English and there are many typos.

Response:

Thanks for your careful reviewing. We have investigated the relevant professional literature and carefully checked the professional words expressed in the manuscript. The modified contents were highlighted with bright yellow background.

Comments:

  1. Most of the content has already come from other papers, and there is nothing special about the paper itself.

Response:

Thanks for your professional comments. As typical carbon-based and ferroelectric materials, various graphene/PZT composite structures were applied for electrical or optical applications. In our paper, we have used the same process to prepared the graphene/PZT composite structure, and some properties including electrical, mechanical and interfacial regulation effects were systematically studied. It may be a good supplement and reference work for brief reading and understanding relevant knowledge. It can also provide some references for the subsequent experimental design.

Reviewer 2 Report

Comments and Suggestions for Authors

Author Response

Response Letter

Reviewer #2:

Comments:

It is interesting to see the results of controlling the domain distribution of PZT by applying PFM probes and bending deformations to achieve p/n doping into graphene and the rectification effect. However, due to insufficient data presented in the paper, I could not determine whether the results shown in Figs. 7 and 8 are valid or not. For example,

Response:

Thanks for your professional comments. We have solved the problems you mentioned one by one.

Comments:

(1) There are no sample photographs.

Response:

Thanks for your comments. We have added sample photographs in figure 7 and figure 8, respectively. The more complete images are shown below:

Figure7. (a) Photograph of graphene/PZT sample on SiO2/Si substrate. The upper right image is a partial enlargement of the electrode gap area. (b) The gray picture displayed the structure topography based on PFM (Piezo-response force microscopy) equipment. The yellow picture shows the domain structure characterization of the middle region of the electrode. (c) Raman spectrum of pure graphene and graphene/PZT integrated film; (b) Test and structure schematic diagram of graphene/PZT device; (c) I-V output electrical transport test curve.

Figure8. (a) Photograph of graphene/PZT sample on mica substrate. Copper electrode with 300 μm thickness was deposited on surface of PZT film by magnetron sputtering equipment. The upper right image is sample image under bending condition. (b) Working diagram of graphene/PZT device; (c, d) I-V output electrical transport test curve of 1 µm and 100 nm thickness respectively.

We also added description of figure 7a, figure 7b and figure 8a in manuscript. The new additions have been marked in bright yellow background.

Comments:

(2) No information on the electrode material of the device (Figs. 7(b) and 8(a) shows "metal film")

Response:

Thanks for your careful reviewing. Three types metal film was adopted in device fabrication process. For graphene/PZT film on SiO2/Si substrate, thinner Ti/Au film was deposited on surface of PZT film to meet the basic electrical conductivity requirement. For graphene/PZT film on mica substrate, thicker metal electrodes were needed to ensure good electrical conductivity in large deformation state. The copper film deposition process based on magnetron sputtering has the advantages of low-cost and fine ductility, then 300 μm thick copper electrode was deposited on surface of PZT/mica suspended structure.

We have added relevant information in figure 1 section of manuscript.

Comments:

(3) No information on device dimensions (distance between electrodes, area of graphene, or he like.)

Response:

The graphene/PZT on SiO2/Si substrate was shown in figure 7a. Ti/Au electrodes were deposited by electron beam evaporation equipment. Graphene with ~1 mm  1 mm size was transferred onto the electrode gap with 10 μm intervals. Different oriented domain structure can be written by PFM probe with the help of external electric field. An obvious dividing line can be found in out-of-plane phase picture of graphene/PZT sample, which means different polarized electric fields were formed to tune carrier density of graphene. For the upper half region, electrons maybe bound to the surface of positive electric field, and more hole carriers were used for transport (p-doping). For the bottom half region, holes maybe bound to the surface of negative electric field, and more electrons were used for transport (n-doping).

The image of graphene/PZT on mica substrate was provided in figure 8a. Thicker copper electrode with 300 μm thickness was prepared by magnetron sputtering equipment. CVD graphene was coated on both sides of PZT film based on wet transfer process. Then the composite structure can be placed on a fixing bracket. Mechanical deformation will be produced when subjected to external pressure, and strain can slightly change domain orientation. The calculated bending radius was ~9.6 mm and ~7.8 mm in electrical test process. Strain gradient maybe induce the change of domain polarization orientation. Then p/n-doping will lead to weak p-n junction inside the graphene film. Comparing I-V curve of graphene/PZT on Si-based substrate and mica substrate, we can find that strain gradient induced surface potential change is weaker than PFM probe modulation method.

We have added relevant information in figure 7 and figure 8 section, respectively.

Comments:

(4) Domain distribution in the experiment shown in Fig. 7 is unknown.

Response:

Thanks for your professional comments. We have added domain distribution image of sample based on PFM probe characterization. The corresponding domain distribution description was also provided in question 3.

Comments:

(5) The size, thickness, and degree of bending deformation of the PZT/Mica sample are not shown. For these reasons, it is difficult to verify the validity of the experimental method and results, and therefore, I could not recommend accepting this manuscript in its present form. My additional comments and questions are shown below.

Response:

Thanks for your professional comments. We have added domain distribution image of sample based on PFM probe characterization. The corresponding domain distribution description was also provided in question 3.

Comments:

  1. In this study, the internal strain ε was calculated using the equation ε=βcosθ/4. Could the authors explain from what theory this equation is derived? Is this different from the Williamson-Hall Plot (WHP) method often used for strain evaluation?

Response:

Thanks for your professional comments. The equation ε=βcosθ/4 is derived from Debye-Scherrer theory, in which ε is calculated mainly by powder diffraction angle. The Williamsen-Hall equation can be reduced to the Scherrer equation when setting the strain eta to zero. Larger strain could broaden the XRD peak, which is more suited for crystalline parameter calculation. PZT film with micro/nano-meter size prepared by sol-gel method has only small strain distribution, then Debye-Scherrer theory maybe more common. (International Journal of Advanced Research in Physical Science, 2015, 2349-7882; Materials Science and Engineering B 120 (2005) 175-180)

Comments:

  1. Were the annealing conditions the same for PZT on Pt/Ti and PZT on mica? Figs. 2(c) and 2(d) show that the thicknesses of PZT on Pt/Ti and PZT on mica were different. Was the thickness intentionally changed in this study? Or is it because it is difficult to control the thickness in this research process?

Response:

Thanks for your comments. The same annealing conditions were adopted for PZT on Pt/Ti and mica. Due to Pt/Ti metal film was protected on Si-based substrate and upper PZT film, metal layer still can maintain good conductivity and uniform morphology.

In order to reduce the leakage current in the domain writing process with the help of PFM-probe, thicker PZT film was prepared on the surface of Pt/Ti metal film. Then it is intentionally changed in figs 2(c) and 2(d). Actually, the thickness of PZT film prepared by sol-gel method can be tuned by changing spin-coating speed of precursor solution. The method is indeed prone to produce thickness deviation.

Comments:

  1. Does the graphene shown by the black line in Fig. 7(c) represent the measured results of the graphene/PZT sample before tuning the domain?

Response:

Thanks for your professional comments. The black and red lines in Fig. 7(c) represent the measured results before and after tuning domain structure in PZT film. The initial domain structure is randomly distribution, and the electrical modulation of graphene by PZT film is unordered. When domain in PZT film was tuned by PFM probe, there are two main types of oriented domain structures are presented in the PZT film. Then the modulated two regions also show obvious p and n doping characteristics. The DC (direct current) output curve exhibit obvious rectification property.

Comments:

  1. The authors should explain the relationship between the polarization state of PZT film and then/p-doping state of graphene and show the n/p-doping regions of graphene in the fabricated device.

Response:

Thanks for your professional comments. There are two types of structures in manuscript has exhibited p/n doping effect and rectification property in electrical curves. We should explain it from two aspects in following discussion. Just as shown in figure 7a and 7b, different oriented domain structure can be written by PFM probe with the help of external electric field. Two oriented domain structures were presented in out-of-plane phase picture. As the result of that, different polarized electric fields were formed on surface of PZT film. Graphene is easily to be tuned by substrate electrical field, so p-type and n-type doping was formed in positive electrical field area and negative electrical field area, respectively. Just as shown in figure 8a and 8b, the graphene/PZT on mica substrate sample was placed on a fixing bracket. When the sample subjected to external pressure, the upper and lower surfaces will produce tensile and compressive forces. For 100 nm thick PZT film, it will produce larger strain gradient to induce the change of domain orientation. Then different oriented domain can be generated on the two sides of PZT film. Considering graphene is easily to be influenced by substrate potential and carriers, p/n doping were formed in graphene coated on the two sides of PZT film.

Comments:

  1. The authors should explain why no rectification effect due to bending deformation was observed in the one μm thick PZT film, but only a small rectification effect was observed in the 100 nm thick PZT film.

Response:

Thanks for your professional comments. Generally, domain structure in ferroelectric film was mainly tuned by external polarized electric field. With the improvement of observation equipment and micro/nano-fabrication technology, the study of strain modulation is becoming a research hotspot gradually. The larger strain gradient maybe promote the domain conversion for further polarization electric field application. In the same bending radius, the PZT film with 100 nm thickness will present more obvious strain gradient and p/n-doping effect. Then a slight rectification characteristic curve is presented in figure. For 1 μm thick PZT film, the surface chargers also can be injected into graphene to enhance the electric conductivity. However, thicker film is difficult to produce enough strain gradient for inducing domain orientation changes. (Nature communication,2020, 3141; Journal of the American Ceramic society, 2018;101:4783–4790)

Comments:

  1. The authors should explain why the I-V curve of the 100-nm-thick sample was nonlinear in the strain-free state.

Response:

Thanks for your careful and professional comments. Comparing with thicker PZT film, the graphene on thinner PZT film is easier to produce a nonlinear output curve in the experimental tests. But we have not found similar tests and analyses in other literatures. The nonlinear phenomenon in figure 8d may be results from electrode contact or influence of PZT film deformation. PZT film with 100 nm thickness is easier to produce domain orientation change, which disturbed the carrier transport in channel. Then higher bias voltage is useful for driving the flow of carriers between electrodes, which lead to higher voltage produce higher current. In other aspects, graphene coated on surface of PZT and electrodes may produce larger wrinkle and slippage, which lead to changes in electrode change and electrical output. Although electrode contact was often used to explain the nonlinear electrical transport properties. It may be not suitable for explaining nonlinear electrical transport in this work.

Round 2

Reviewer 1 Report

Comments and Suggestions for Authors

Thanks for your good response. I will accept. 

Comments on the Quality of English Language

No comment.

Reviewer 2 Report

Comments and Suggestions for Authors

As shown by the application of graphene to strain gauges, there is a strain dependence in the I-V characteristics of graphene. In addition, the I-V characteristics of graphene devices change due to strain-induced changes in the physical and electrical contact between graphene and metal electrodes, as well as changes in the wrinkle state of graphene. In this paper, the authors reported that the p/n doping effect due to strain loading was not observed in the 1-μm-thick sample, whereas the rectification effect due to strain loading was observed in the 100-nm-thick sample. However, since strain due to bending deformation increases with film thickness, if the rectification effect is due to the p/n doping effect by strain loading, the strain effect should be more significant for the sample with a film thickness of 1 µm. Therefore, from the results in Fig. 8, it cannot be determined whether the slight rectification property observed in the 100 nm-thick sample is due to the polarity modulation of the PZT film or simply to the strain dependence of the I-V characteristics of the graphene device. If the authors intend to claim that the slightly observed rectification properties are due to modulation of the polarization properties of PZT due to bending deformation, they at least need to show the change in the electrical polarization of PZT under the bending deformation in this study.

In response to Reviewer 1's comment that "Most of the content has already come from other papers, and there is nothing special about the paper itself.", specific differences or uniqueness must be shown.
